# Assessing the Spatial Distribution Pattern of Street Greenery and Its Relationship with Socioeconomic Status and the Built Environment in Shanghai, China

**Chao Xiao, Qian Shi and Chen-Jie Gu \***

School Economics and Management, Tongji University, Shanghai 200092, China; chaoxiao@tongji.edu.cn (C.X.); qianshi@tongji.edu.cn (Q.S.)
**\*** Correspondence: 1910475@tongji.edu.cn

**Abstract:** Urban greenery is widely acknowledged as a key element for creating livable urban environments and improving residents' quality of life. However, only a few current studies on the subject of urban greenery focus on a human visual perspective and take street greenery into consideration. Street greenery is an indispensable component of urban vegetation to which residents have a higher frequency of access. Additionally, few studies focused on the disparity of the green view at a micro-level, such as at a county or community level. This study explored the spatial distribution of street greenery and its influential factors using the green view index (GVI) as the main evaluation indicator. Compared to other traditional indicators of greenery, such as the normalized difference vegetation index (NDVI) and accessibility, GVI is recognized as a human-oriented indicator to evaluate the quantity of greenery viewed by human eyes in daily life. The downtown area of Shanghai was chosen as the case study, as it reflects the common phenomenon of street greenery in many megacities globally. In addition, county/jiedao (the same administrative area as county in China) level was selected as the minimum geographical unit to evaluate the disparity of GVI and its influential factors to fill the knowledge gap. We analyzed 233,000 pieces of street-view images from Baidu Map and other correlated data. The results showed (1) the street greenery of 70% of the downtown area of Shanghai is less than the recommended comforFogre visual environment; (2) street greenery is spatially clustered in Huangpu district, Xuhui district, college town, and the Century Park of Shanghai; (3) street-greenery distribution is positively correlated with housing price and street network density, and negatively correlated with the ratio of society vulnerability; however, it is uncorrelated to population density. According to these findings, local municipalities could improve urban planning and design by introducing a more human-oriented green-space policy that improves social equity.

**Keywords:** street greenery; green view index; street-view image; social equity

## 1. Introduction

With the high-speed process of urbanization, both large urban populations and expanding urban areas lead to environmental degradation. Environmental problems caused by the impact of human disturbances on ecosystems reduce the benefits of ecosystem services for human wellbeing, resulting in both physical and psychological issues [1,2]. Lessons drawn from London in the 1950s and Beijing in the 2010s highlight that the living environment is vitally important for enhancing human happiness [3,4]. As urbanization continues, urban greenery, including trees, shrubs, forests, and all kinds of vegetation, and the ecological services that it offers, are widely acknowledged as key elements for creating livable urban environments and improving residents' quality of life [5–7]. Ecosystem services provided by urban greenery include air and water purification [8], weakening urban-heat-island effects [9], enhancing biodiversity [10], and mitigating natural disasters [11]. Residents who have more opportunities to access and undertake physical and

social activities significantly enhance their physical and mental health [12,13]. Hence, with the purpose of maximizing the wellbeing provided by urban green spaces, the space layout of urban greenery should be rational and considered in spatial planning for each social group.

In spite of the large number of benefits from urban vegetation, there is evidence that the spatial distribution of urban greenery is inadequate and unjust in cities, which indicates that some residents are deprived of receiving these ecological services and benefits provided by urban vegetation [14–16]. Some scholars indicated that environmental injustice is inherently exclusive to wellbeing, including the physical and mental health of vulnerable communities [17]. These communities also suffer from a disproportionate amount of public services, such as access to urban green spaces. Numerous studies show that racial minorities, low-income groups, and vulnerable groups have inequitably low access to urban green spaces compared to that of affluent and privileged groups [18,19]. For example, a study in Latin America showed that accessibility to adjacent greenery for vulnerable groups, including the disabled, low-income groups, and residents in suburban areas is much lower [20]. Venter et al. found out that green spaces in high-income communities or advantaged racial communities are much more abundant, accessible, and high-quality in South Africa [21]. In addition, Nesbitt et al. concluded that social groups with higher-education degrees and higher incomes have more access to urban vegetation in 10 US cities [22]. Another study in Nanjing, China, revealed that the accessibility of urban parks is positively related to housing prices and negatively related to the age of buildings [23]. According to the former studies, urban parks, urban green spaces, and street greenery, which might be accessible more frequently in daily life, should be focused on to evaluate the equity of its spatial distribution. When moving beyond environmental inequalities, environmental injustice might take social and political claims into account, since this is what the policy makers need to consider [24].

Current evaluations of urban greenery are mainly based on the simplistic availability of urban green spaces by calculating the area amount, convenience of access to urban green spaces by calculating the distance or canopy cover calculated by satellite images [22,25–27], and few studies focused on urban greenery from a human visual perspective. These studies provided systematic quantitative assessment of urban greenery, but there are some drawbacks, for instance, data accuracy on the district scale, and the rationality of 'accessibility'. With regard to the accuracy of data and results on the district scale, the availability of urban green spaces only evaluates the overall or average ratio of green coverage on a city or larger scale. At a microlevel, such as the county or community level, disparity and the specific spatial distribution of green spaces are not taken into account [28]. Green coverage derived from high-resolution satellite images could be useful at a smaller scale, such as city- or nationwide. However, from a human-oriented and a more microperspective, satellite images are unable to provide accurate spatial layout of street greenery [29]. In addition, one of the most widely used method of accessibility only takes visiting distance as the measurement of proximity to a green space [30], but neglects the frequency and potential for different social groups to access green spaces, since the vast majority of people work or study during weekdays [31]. Selecting distance or time cost as the evaluation criterion of accessibility fails to take the travel behaviors of residents into consideration, since different behaviors may lead to different accessibility to urban greenery.

To overcome these barriers, human visually perceived greenery from a three-dimensional perspective has attracted scientific and public attention, as it is different from the arterial view images, and visually reflect residents' perception of urban greenery [32]. This does not necessarily require residents to be fully immersed in urban green spaces; residents' visibility of street greenery while walking can also have positive mental effects, which is often overlooked in current studies of urban greenery. Furthermore, from the perspective of policy makers and urban planners, GVI is able to help them understand residents'

real demands on green space, since it reflects real feelings of greenery from residents' visual system.

Urban street greenery, as an indispensable component of urban vegetation, including trees, lawns, and shrubs along street networks, plays an important role in attracting residents and improving the walkability of streets [33]. It can also mitigate visual intrusions caused by local transportation and improve the aesthetic value of a city [34]. Studies indicated that more visible greenery could obtain stronger public support than less visible greenery can in the case of the same urban green coverage [35]. The visibility of greenery contributes to the wellbeing of residents in their living environments. According to Aoki's study [36] on green view (GV), 30% GV should be the standard of a comfortable visual environment when walking on streets as a measurement of the quality of street greenery. Hence, a more objective and human-oriented analytical method for residents' visibility of street greenery should be proposed as a necessary supplement of urban greenery study. Yang et al. adopted pictures to evaluate visual greenery as representative of people's view of the amount of urban green, which is an effective and efficient green metric called the green view index (GVI) [35]. To overcome the time-consuming process of data collection through manual photography, Li et al. developed a GVI method based on Google Street View images that can calculate large numbers of images on a large scale [29].

The aim of this paper is to adopt a BaiduStreetMap-based GVI as the intuitive method to calculate street greenery from residents' perceptions. The authors also explored how unequal the distribution pattern of the green view is, and its inter-relation with the social status of the urban population and built environment based on a street-greenery analysis of the downtown area of Shanghai at a county or jiedao (the same administrative area as county in China) level. Through this analysis, the local municipality may have references to improve the spatial layout of street greenery beyond the distribution inequalities of green spaces, promote equity among different social groups, and provide guidance for better spatial planning.

## 2. Methodology

### 2.1. Study Area

Shanghai, as an international megacity in China that contains a large number of different social groups, was selected as an empirical case study that reflects the common example of street greenery in many megacities globally. Urban-regeneration activities are conducted in Shanghai to improve the quality of both gray and green infrastructure, including urban green spaces. The geographic location of Shanghai is between $120°52'$ to $122°12'$ east longitude and $30°40'$ to $31°53'$ north latitude, and the total area of Shanghai is 6340 km$^2$. Shanghai is the most developed city in China, with a total GDP of CNY 3870 billion and a total population of 24,280,000 [37]. The study area was downtown Shanghai (Figure 1), surrounded by the outer ring expressway (boundary of the study area) with a total of about 660 km$^2$. The majority of economic and social activities in Shanghai are developed in the downtown area in the outer ring. Counties in the study area suffer from unbalanced economic growth and public resources, including urban greenery; thus, ensuring urban green equity for all social groups in different counties is the target of the local municipalities.

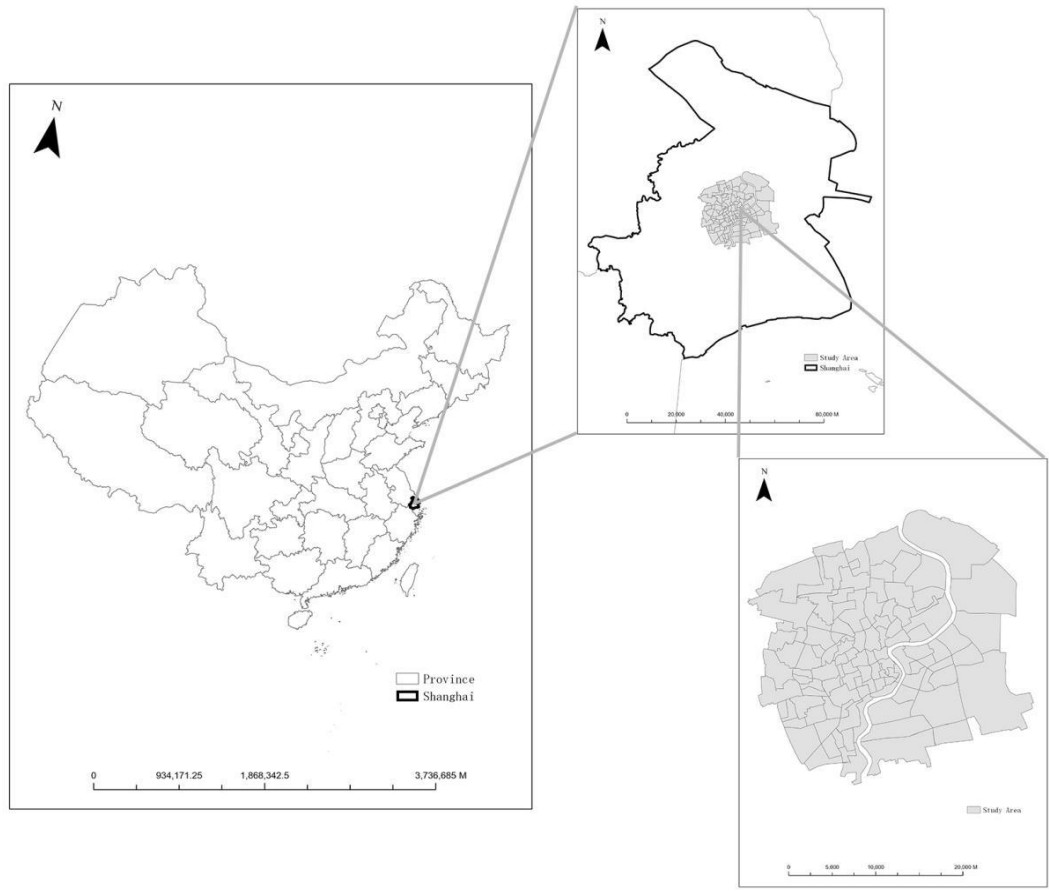

**Figure 1.** Study area.

## 2.2. Data and Processing

The geographical coordinate data of sampling points were extracted from an Open-StreetMap dataset of the street network of Shanghai (Supplementary Materials). Figure 2 is the holistic street network of the study area that shows the use of ArcGIS to convert the street network into the sampling points with an interval of 100 m along each street in the downtown area of Shanghai (using Figure 2c as a zooming example to see the sampling points alongside the street network). We obtained 66,160 sampling points covering the holistic study area. Then, according to the geographical coordinates of these sampling points, a 360° panoramic camera from Baidu Street Map offered four street-view images at four directions, including angles of 0°, 90°, 180°, and 270° at each point to form a panoramic view (Figure 3). We used Python to set up the application program interface (API) keys applied from BaiduMap with the parameters of field of view (FOV; 0°, 90°, 180°, and 270°), and the width and height of the picture (1024 × 512 pixels) to download the street-view images. The requested link is http://api.map.baidu.com/panorama/v2?ak=E4805d16520de693a3fe707cdc962045&width=1024&height=512&location=121.546400333912,31.2269847559773&fov=0,90,180,270 (accessed on 18 August 2021). Figure 3 shows four sample street-view images downloaded according to the geographical coordinates of a sampling point. The camera was installed on top of the Baidu Map car at a height of about 1.6 m, suitably representing people's eye level [38]. Excluding the invalid access of the geographical-coordinate information of some sampling points; 58,309 sample sites and about 233,000 pieces of street-view images were crawled from BaiduMap in total.

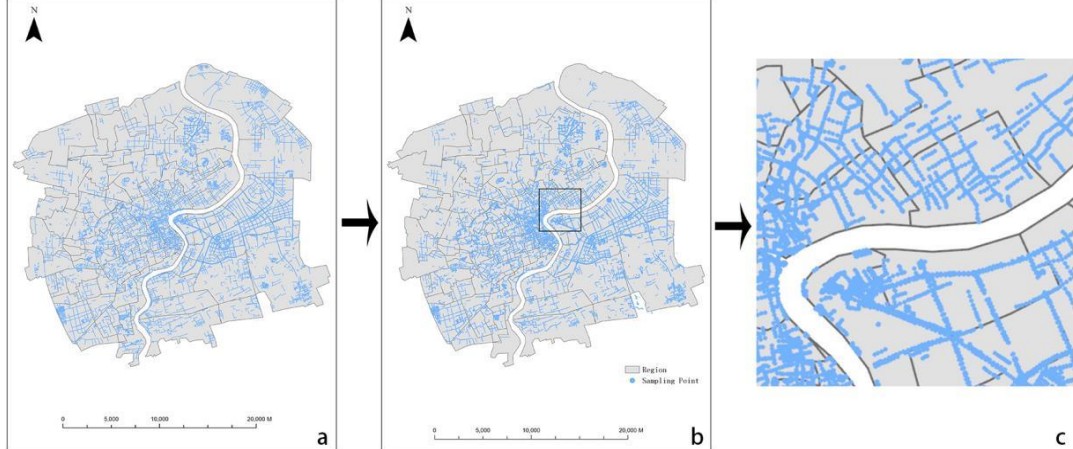

**Figure 2.** Street network and sampling points. (**a**) holistic street network of the study area; (**b**) sampling points; (**c**) zooming example.

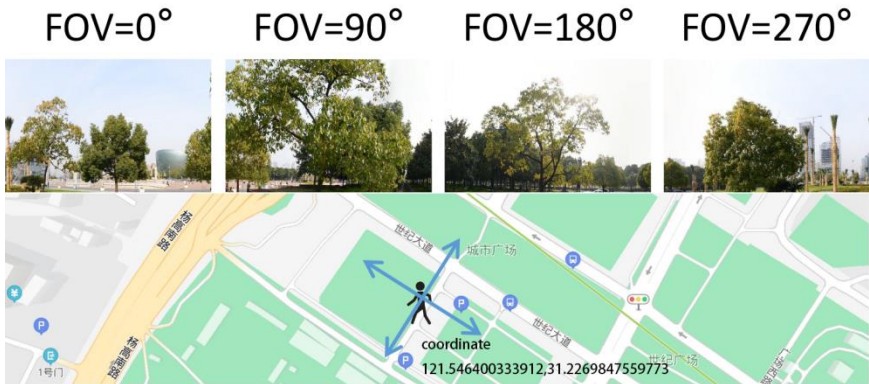

**Figure 3.** Four street-view images from four directions at one sampling point.

## 2.3. Green View Index (GVI) Calculation

Aoki first proposed the indicator of greenery or green view as the percentage of greenness perceived by people's visuality [36]. Yang et al. defined green view as the amount of greenness that people can see from their perspective [35]. According to these studies, we used GVI to calculate the greenery of each street view image to represent the amount of greenery visible to the human eyes. We converted the format of each image from RGB to HSV, and extracted the value of the hue channel. The HSV model of the picture, from 0° going counterclockwise to 360°, represents red (0°), green (120°), and blue (240°) (Figure 4). On the basis of the color spectrum, the value of the hue channel in the OpenCV package in Python ranged from 0 to 180, half of 0 to 360 of the HSV color model; hence, we defined the value of the hue channel, which ranged from 35 to 77, as green to represent greenery at streets (Table 1). Using the color range to represent greenness is in accordance with the ground truth recognized by computers.

From each sampling site, we downloaded four test street-view images at four directions, and calculated the percentage of GVI through a recognition algorithm using OpenCV to obtain an average result of GVI for each sampling point (Figure 5). The calculation of GVI is based on the following equation:

$$\text{Green view index} = \frac{\sum_{i=1}^{4} Green\_i}{\sum_{i=1}^{4} Total\_i} \times 100\% \tag{1}$$

where *Green_i* is the number of green pixels in the four images at four directions, and *Total_i* is the pixel number of the four images at four directions.

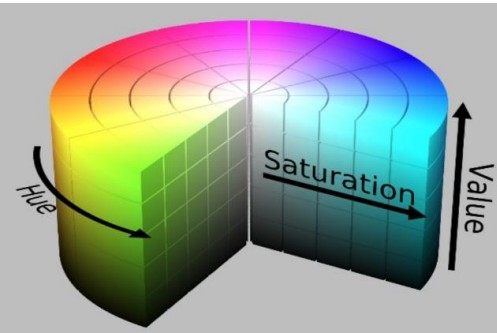

**Figure 4.** Color spectrum in HSV model from: https://my.oschina.net/u/4851203/blog/4825518 (accessed on 18 August 2021).

**Table 1.** Value of hue channel of different color.

|  | Black | Grey | White | Red | | Orange | Yellow | Green | Cyan | Blue | Purple |
|---|---|---|---|---|---|---|---|---|---|---|---|
| H-min | 0 | 0 | 0 | 0 | 156 | 11 | 26 | 35 | 78 | 100 | 125 |
| H-max | 180 | 180 | 180 | 10 | 180 | 25 | 34 | 77 | 99 | 124 | 125 |
| S-min | 0 | 0 | 0 | 43 | | 43 | 43 | 43 | 43 | 43 | 43 |
| S-max | 255 | 43 | 30 | 255 | | 255 | 255 | 255 | 255 | 255 | 255 |
| V-min | 0 | 46 | 221 | 46 | | 46 | 46 | 46 | 46 | 46 | 46 |
| V-max | 46 | 220 | 255 | 255 | | 255 | 255 | 255 | 255 | 255 | 255 |

Adapted from: https://www.cnblogs.com/wangyblzu/p/5710715.html (accessed on 18 August 2021).

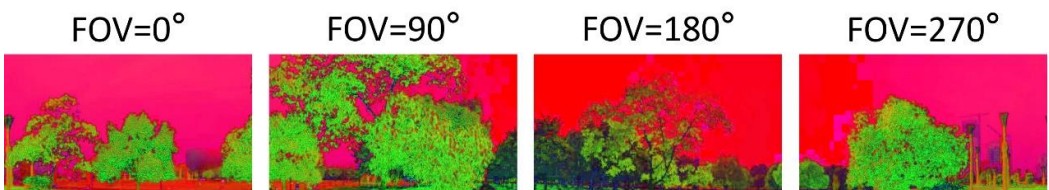

**Figure 5.** Recognition result of each street-view image.

In this study, sampling sites were selected from walkable streets, for example cycleways, footways, residential streets, steps, and tracks, to largely avoid intrusions of nonbotanical green objects such as vehicles and road signs. In order to conduct comprehensive analysis to find the spatial patterns of the GVI and the relationships between the disparity of GVI and other factors, we aggregated the value of the GVI of each point at the county level. Considering the clustering effect of sampling points at the county or jiedao level, the simplistic mean value of these points in a given area might be biased [39]. With the purpose of mitigating the bias of mean values resulting from the clustering effect, the Voronoi tessellation (Thiessen polygon) method was introduced. According to the Thiessen polygon method, each sampling point is associated with only one specific region in a given county, and these specific regions are nonoverlapping, contiguous, and homogeneous [40]. In this study, such a cluster probability model can give weight to each sampling point on the basis of the area of its Thiessen polygon. The equation of the aggregated value of GVI is as follows:

$$\text{Aggregated GVI} = \sum_i^1 GVI\_i \times \frac{A\_i}{A} \tag{2}$$

where $GVI\_i$ is the value of the GVI of point $i$ in the county or jiedao, $A\_i$ is the area of the Thiessen polygon associated with point $I$, and $A$ is the area of the county or jiedao.

For example, Figure 6 shows Jiang Su Lu county (jiedao) as an example to illustrate the Thiessen polygon of each sampling point in this county.

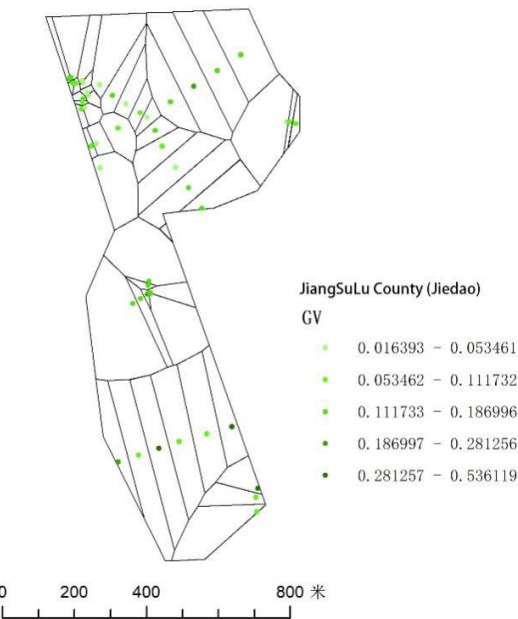

**Figure 6.** Thiessen polygon of each sampling point in Jiang Su Lu county (jiedao).

### 2.4. Socioeconomic and Built Environment Variables

With the purpose of exploring the equity of the spatial distribution of street greenery, we selected two categories of variables, and their data are accessible from census data and other data companies (Table 2). On the basis of previous social-justice and environmental-equity studies, one category is socioeconomic variables, including population density, ratio of vulnerable population, and housing price [14,41]. Since the studied county level is a comparatively microscale of urban areas, limited socioeconomic data could be obtained from the Internet. Among socioeconomic variables, population density and the ratio of vulnerable groups on the county or jiedao level were extracted and preprocessed from the sixth nationwide census data [42], in which other indicators such as gender, educational background, and income were recorded as more large-scale aggregate data on the city level. In addition, housing-price data at the county or jiedao level are useful to more accurately evaluate the economic status of residents [43]; in this study, data were crawled from LianJia, a well-known real-estate agent in China. The built-environment variable may directly impact the spatial distribution of the GVI through the capacity for constructing street greenery, including point-of-interest (POI) diversity and street-network density [14,44]. Through calculations via the street length and area of the county, the density of the street network determines the space of street greenery in a given block, especially walkable streets for pedestrians, since the two sides of expressways are usually guardrails. POI diversity to some extent determines the degree of vitality and development of the block, which might be an interactional factor with the GVI in that both land-use diversity and the construction of green spaces are managed by local municipalities in China.

**Table 2.** Selected variables.

| Variables | Explanations | Data Sources | Reference |
| --- | --- | --- | --- |
| Population density | Density of population in each unit | Calculated by census data | Landry and Chakraborty, 2009 |
| Ratio of vulnerable group | Ratio of people younger than 16 and older than 60 | Calculated by census data | Landry and Chakraborty, 2009 |
| Housing price | Average price of house in each unit | Lianjia | Li et al., 2019 |
| POI diversity | Diversity of point of interest | BaiduMap | Grove et al., 2006 |
| Street network density | Density of walkable streets | OpenStreetMap | Grove et al., 2006 |

The spatial patterns and cartographic visualization of the GVI were analyzed in ArcGIS 10.2. Descriptive statistical analysis of the GVI and correlations between the variables and the GVI at a county or jiedao level were explored through Spearman's correlation analysis with SPSS Statistic 25 (n = 58,309 for the value of GVI of sampling sites; n = 111 for analyzing county-level GVI and other variables).

## 3. Results

### 3.1. General Overview of Current Spatial Pattern of Street Greenery in Shanghai

This study first explored the current situation of street-greenery patterns in Shanghai through descriptive statistical analysis of the GVI value in the downtown area. GVI values among all sampling points ranged from 0.029% to 87.2% (Figure 7), and the mean value was 22.2%. The standard GVI deviation of 17.4% showed a considerable gap between green and nongreen sampling sites. According to Table 3 and Figure 8a, 32.4% of the GVI values were under 0.1, and about 55% of the values were lower than 0.2. About two-thirds of the GVI values were lower than 0.25, which means that the amount of street greenery in 2/3 of the downtown area of Shanghai failed to live up to the standard [45]. Points with high GVI values are usually located south-west and south-east of the downtown area. Approximately 30% of the GVI values were higher than 0.3, which indicated that 70% of the sampling sites are unable to provide a comfortable visual perception based on the recommended GVI of 30% proposed by Aoki [45]. Hence, from the perspective of green views, there is much room for improving both the quality and quantity of urban green spaces of the downtown area in Shanghai, for example through transforming parts of gated urban parks or large areas of green spaces into street greenery or open green spaces alongside streets to narrow the gap of GVI values of each unit. In addition, the local municipality may focus more on visible green spaces in counties adjacent to the outer ring, in which land characteristics are usually industrial. North-west and north-east of the study area, GVI values were much lower than those of the central and southern urban areas. These industrial land areas usually have large numbers of buildings, warehouses, and factories with bare streets without plants and trees.

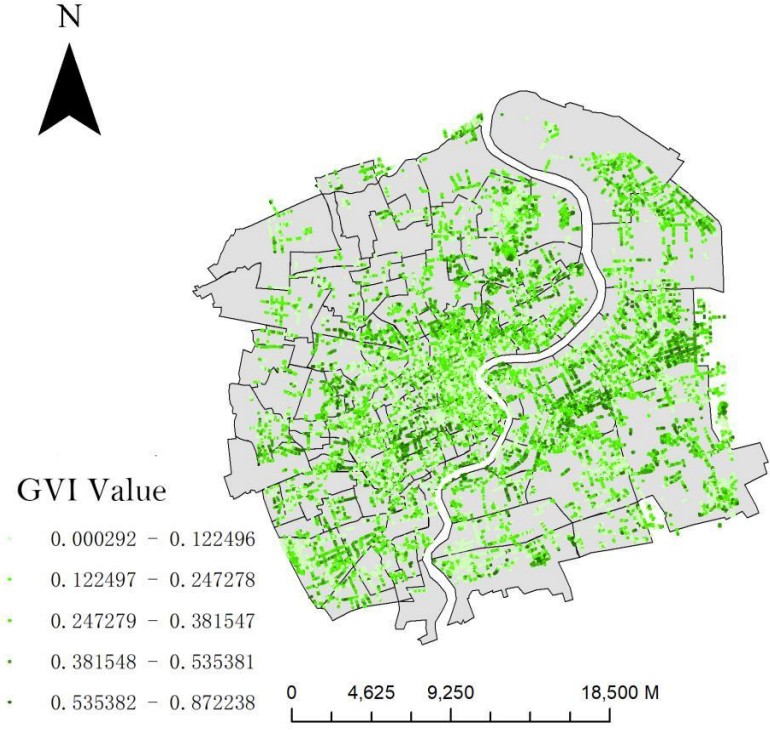

**Figure 7.** Green-view value of each sampling point.

**Table 3.** Descriptive statistics of green-view values.

| | N | Min | Max | Mean | Std Dev |
|---|---|---|---|---|---|
| GV | 58,309 | 0.00029 | 0.87223 | 0.222 | 0.174 |
| **Aggregated GVI at County Level** | 111 | 0.101 | 0.378 | 0.198 | 0.560 |

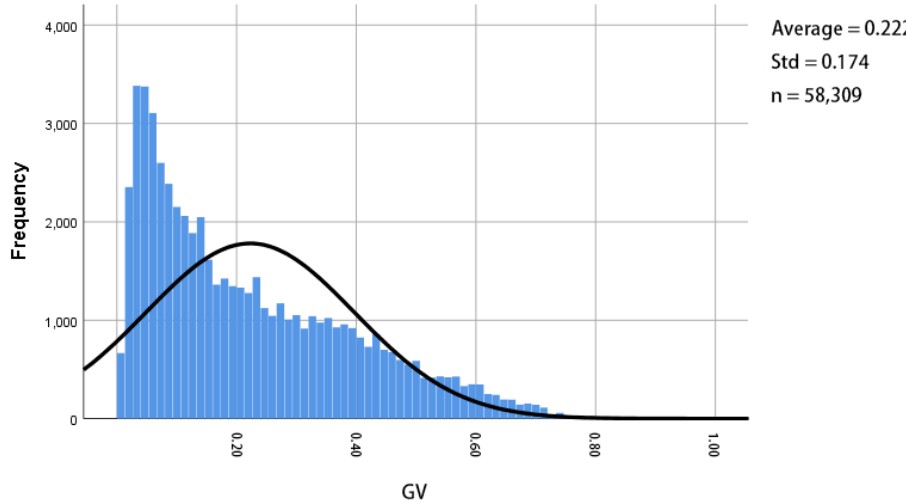

(**a**)

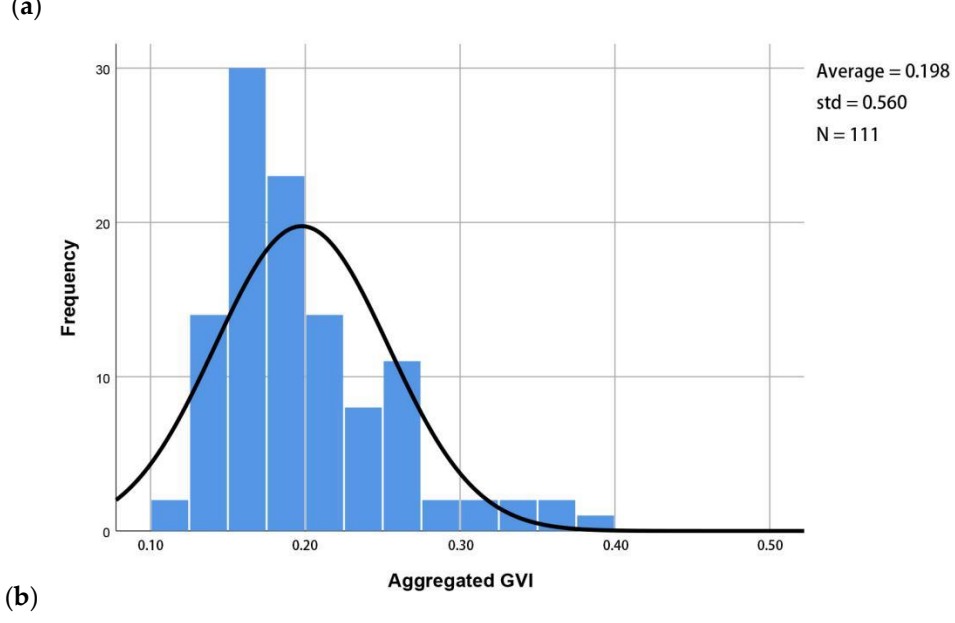

(**b**)

**Figure 8.** (**a**). Descriptive statistics of point green-view values; (**b**) Descriptive statistics of green-view value at county or jiedao level.

### 3.2. Spatial Clustering of Street Greenery in Downtown Area of Shanghai

Although Figure 7 intuitively demonstrates the spatial pattern of point-level GVI values, analysis from the county level was conducted to explore the spatial clustering of GVI. At the county level of the downtown area in Shanghai, the aggregated GVI value varied from 10.1% to 37.8%, and the standard deviation was 0.56 (Figure 8b and Table 3), which indicated that spatial patterns were more balanced than those of the sampling sites. Figure 9 demonstrates that the spatially clustered effect of the GVI values at the county level was still obvious according to Moran I's score (Moran I score = 0.31, *p* value < 0.01). The matrix of the spatial weights in this study was generated by Queen's Case, which

means that two polygons are adjacent if they have sharing points or edges. Results of Moran I analysis indicated that it is possible for counties or jiedaos with higher GVI values to be adjacent to those with similarly higher GVI values. Counties or jiedaos with low GVI values, on the other hand, are usually clustered together. The majority of the top ten counties or jiedaos with the highest GVI values are in the old central urban areas in inner ring districts (central urban areas), especially along the waterfront belt of Huangpu River and southwest of the inner ring districts (Figure 10). These two clusters have large numbers of landmarks and plants along the streets, among which are the famous Chinese parasol trees, contributing to the city image of Shanghai; for example, counties or jiedaos in the Huangpu and Xuhui districts, south-west of the inner ring districts. According to the local municipality of Huangpu, landscape planning focused on green belts and open micro-urban parks to improve residents' feelings of greenery [46]. In addition, the local municipality of Xuhui proposed that smaller and microgreen spaces and pocket parks should be constructed to fill gaps in street greenery [47]. The remaining top ten counties or jiedaos with the highest GV values were located outside the inner ring and are surrounded by large green spaces such as parks or waterfront areas, especially jiedaos adjacent to Pudong Century Park, Fudan University, and Tongji University at College Town. Pudong Century Park is the largest eco-park in the inner ring of Shanghai with a large amount of grassland, plants, and trees at the boundary of the park offering a high value of GVI. Wujiaochang jiedao, named College Town, is centered on Fudan and Tongji Universities, and was constructed as the central intelligence district (CID) in Shanghai [48]. The environment of the Wujiaochang jiedao has walkable streets and abundant public green spaces because of the policy of constructing a high-quality environment to attract more talent.

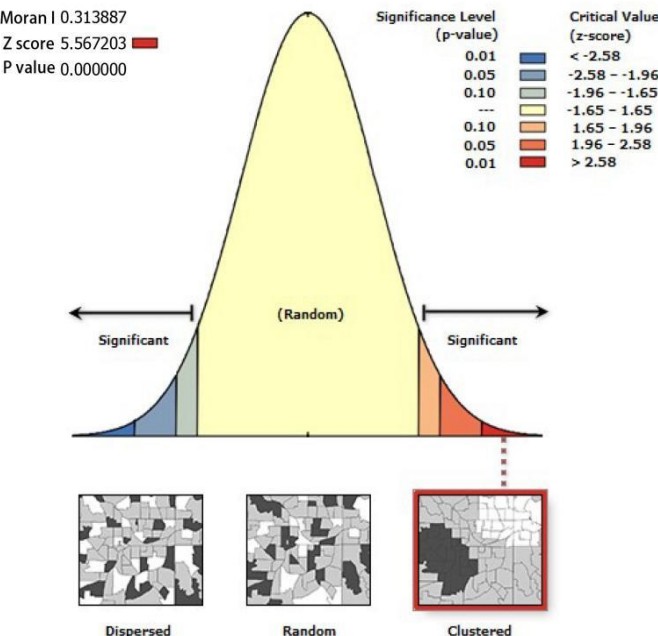

**Figure 9.** Spatial-clustering effect of green-view value.

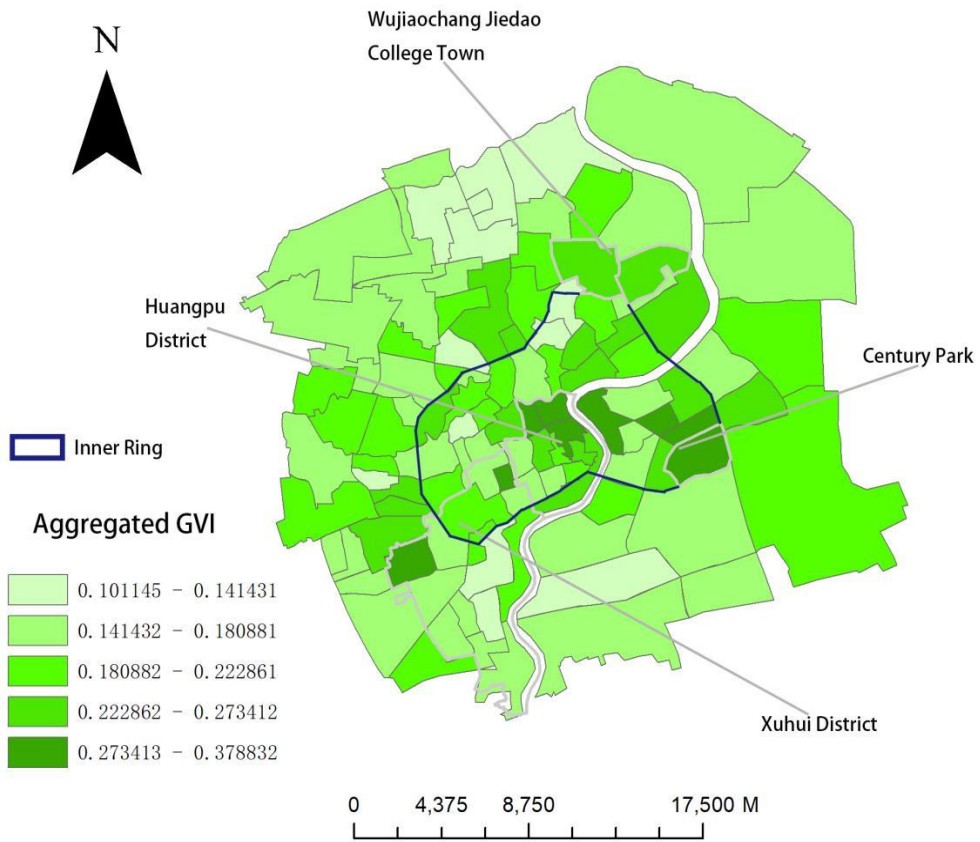

**Figure 10.** Aggregated green-view value at county or jiedao level.

*3.3. Correlation between Street Greenery, and Socioeconomic and Built-Environment Variables*

In order to explore the equity and factors of the spatial clustering of GVI values, this study analyzes the correlation between GVI values and selected variables. Table 4 shows the outcome of Spearman's correlation analysis between GVI values of the county or jiedao, and the selected socioeconomic and built-environment indicators. Table 4 and Figure 11 show that GVI values are positively correlated with economic and built-environment indicators, including housing price, POI diversity, and street-network density. Counties or jiedaos with a more prosperous economy or a complete built environment thus usually possess higher GVI values, especially those districts with high housing prices and dense walkable street networks. The coefficient between street network density and GVI values was about 0.49 (sig < 0.01). This indicates that street network density, especially the density of walkable streets, positively influences GVI values since a large majority of the sampling points in this study were selected from walkable environments to simulate a view of pedestrians.

**Table 4.** Correlation analysis of green view.

| | | Population Density | Vulnerable Groups | Housing Price | POI Diversity | Street-Network Density |
|---|---|---|---|---|---|---|
| GVI | Spearman Correlation | −0.42 | −0.172 * | 0.433 ** | 0.356 ** | 0.489 ** |
| | Significance | 0.329 | 0.035 | 0.003 | 0.001 | 0.001 |
| | Number | 111 | 111 | 111 | 111 | 111 |

* significant at the 0.05 level, ** significant at the 0.01 level.

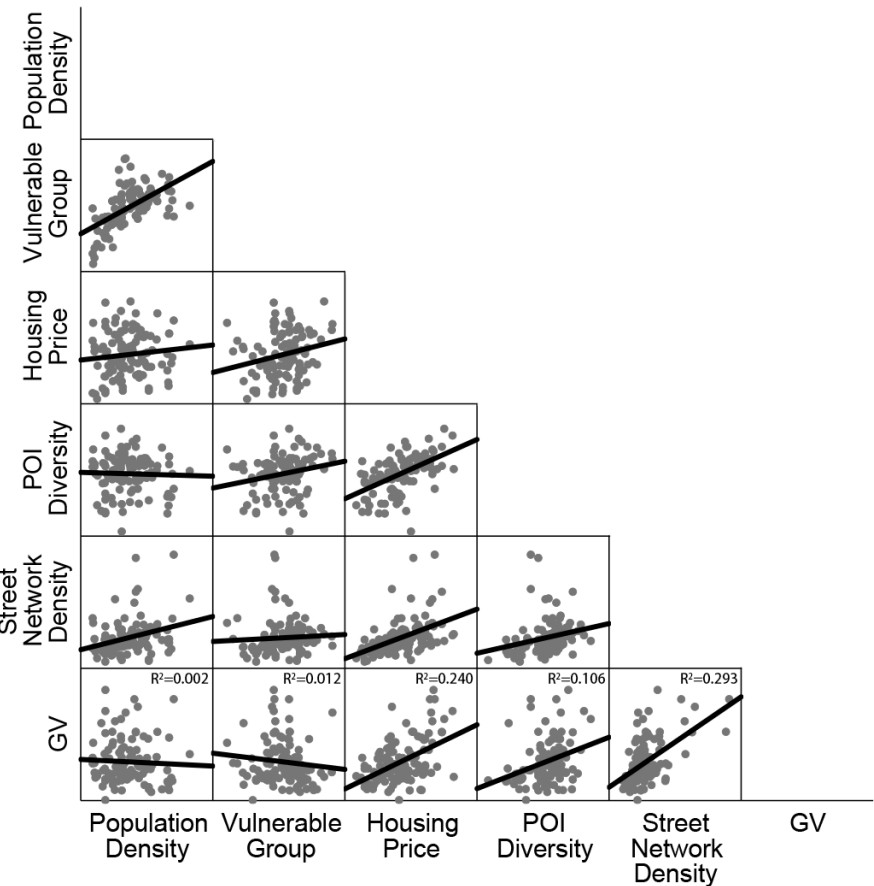

**Figure 11.** Correlation between GVI and other variables.

GVI values were negatively correlated with the category of census data in this study, among which the ratio of vulnerable groups is statistically significant. However, population density is uncorrelated with GVI values (sig > 0.05). The uncorrelated relationship between population density and GVI at county level is inconsistent with the conclusion of some previous studies that population density is positively correlated with GVI at city level [38].This may be because the population is evenly distributed in the inner ring of Shanghai, and several subcenters are also scattered all over the downtown area since residential clusters in Shanghai present a monocentric structure [43]. As a result, the population of Shanghai is also evenly distributed in each subcenter, and spatial GVI clusters are uncorrelated with population density. In addition, the difference between the outcomes of former studies and this study might be caused by the different geographical research unit, and from a more micro level in this study, the uncorrelation between population density and GVI is more reliable.

Ordinary least-squares (OLS) multivariate-regression models were introduced to analyze the interactions between GVI values and these variables. Table 5 illustrates the coefficients, t-values, and $R^2$ of the OLS model. The $R^2$ of 0.42 indicates a comparatively good-fit model. According to the relationship between t-values, statistical significance and degree of freedom (n = 111 in this study), if $|t|$ values > 1.96, it means that the outcome is significant at the 0.05 level, and if $|t|$ values > 2.58, it means that the outcome is significant at the 0.01 level. Two of the three socioeconomic variables were not statistically significant in the OLS model ($|t|$ values < 1.64, not significant), and housing price was positively correlated with GVI values ($|t|$ value > 2.58, significant at the 0.01 level). Concerning the two built-environment variables, POI diversity was not significantly correlated ($|t|$ value < 1.64) with GVI values; however, street network density was more strongly positively correlated with GVI values ($|t|$ value > 2.58, significant at the 0.01 level) since the coefficient was much higher than that of the housing price. However, according to

Moran's I analysis, residuals of the OLS model were strongly spatially dependent, which indicates that the OLS model could not explain all relations from a spatial perspective. Hence, geographically weighted regression (GWR) models were adopted to mitigate the spatial dependence of the residuals of the OLS model. Mean coefficients and $R^2$ are shown in Table 5 (t-values and local $R^2$ of each county or jiedao are in Supplementary Materials). The $R^2$ of 0.45 of the GWR model means that the good model fit was increased. Housing price and street-network density were also significantly positively correlated with GVI values ($|\bar{t}|$ value > 2.58, significant at the 0.01 level), and street-network density has much stronger influence on GVI values than housing price does. In addition, compared to the OLS model, vulnerable groups were significantly negatively correlated with GVI values ($|\bar{t}|$ value > 1.96, significant at the 0.05 level) in the GWR model, which indicates that the negative influences of vulnerable groups are statistically significant when considering the spatial dependence of variables.

**Table 5.** OLS and GWR analyses of green view.

| | OLS | | GWR | |
|---|---|---|---|---|
| | **Coefficient** | **\|$t$\| Value** | **Mean Coefficient** | **\|$\bar{t}$\| Value** |
| Intercept | $8.909 \times 10^{-16}$ | 0.490 | 0.009 | 0.159 |
| Population Density | −0.113 | 1.111 | −0.092 | 0.934 |
| Vulnerable Group | −0.157 | 1.581 | −0.226 * | 2.351 |
| Housing Price | 0.276 ** | 2.800 | 0.290 ** | 2.947 |
| POI Diversity | 0.106 | 1.160 | 0.113 | 1.251 |
| Street-Network Density | 0.428 ** | 4.657 | 0.409 ** | 4.477 |
| $R^2$ | 0.424 | | 0.449 | |

* significant at the 0.05 level, ** significant at the 0.01 level.

Figures 12–14 illustrate the different coefficients of vulnerable groups, housing price, and street-network density at each county or jiedao from the result of GWR model. Figure 12 shows that vulnerable groups were more correlated with GVI values in the inner ring (central urban area) than those east and south-east of the study area. Low GVI values in such counties or jiedaos may be caused by the dense population and dilapidated living environments in the old central area. The young and middle-aged population usually pursue a more comfortable living environment in new communities, and vulnerable groups are often left at resettlement housing after the demolition of their original residences. The distribution of the housing-price coefficient (Figure 13) was relatively strongly correlated with GVI values in central area and the north and east of the study area. The reason might be that large numbers of new upscale communities are being planned and constructed in the north and east of the study area, but other infrastructures such as green spaces and walkable streets have not been allocated at the whole county level. Additionally, local municipalities and real estate only distribute parts of greenery surrounding communities with relatively high housing prices, especially in the central area. Figure 14 displays the spatial layout of the coefficient of street-network density on GVI value at each county or jiedao. The western and southern parts of the study area showed stronger correlation with the GVI value than that east and north of the study area. This may be caused by the urbanization process and policy of Shanghai that local municipalities tended to first develop the southwest of Shanghai. In consequence, street-network density, green spaces, retail, infrastructure, and other high-quality built-environment elements were developed in the southwest of Shanghai. Hence, especially in the western and southern parts of the study area, the more walkable the district is, the higher GVI value the district has.

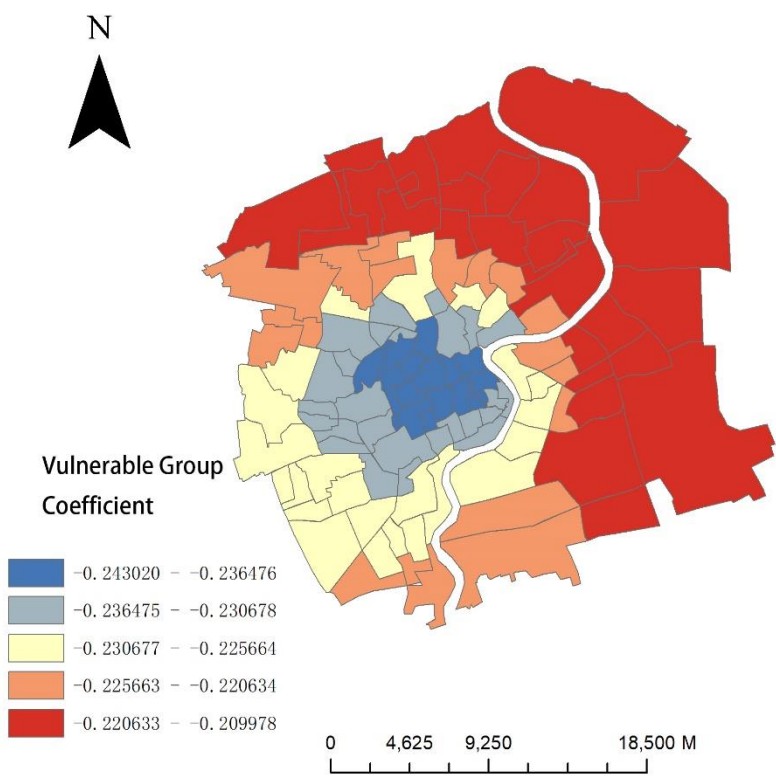

**Figure 12.** Vulnerable-group coefficients at each county or jiedao.

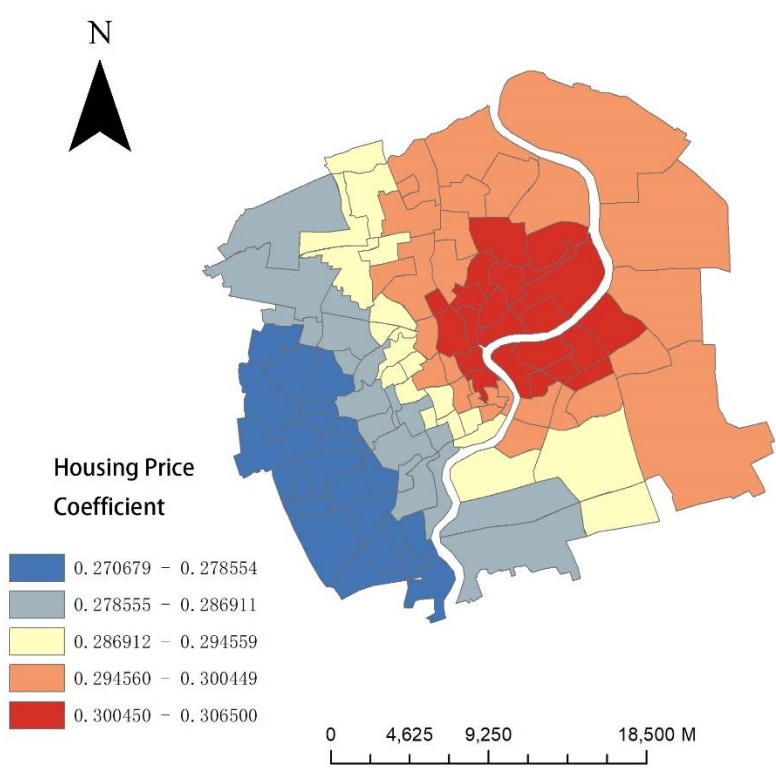

**Figure 13.** Housing-price coefficients at each county or jiedao.

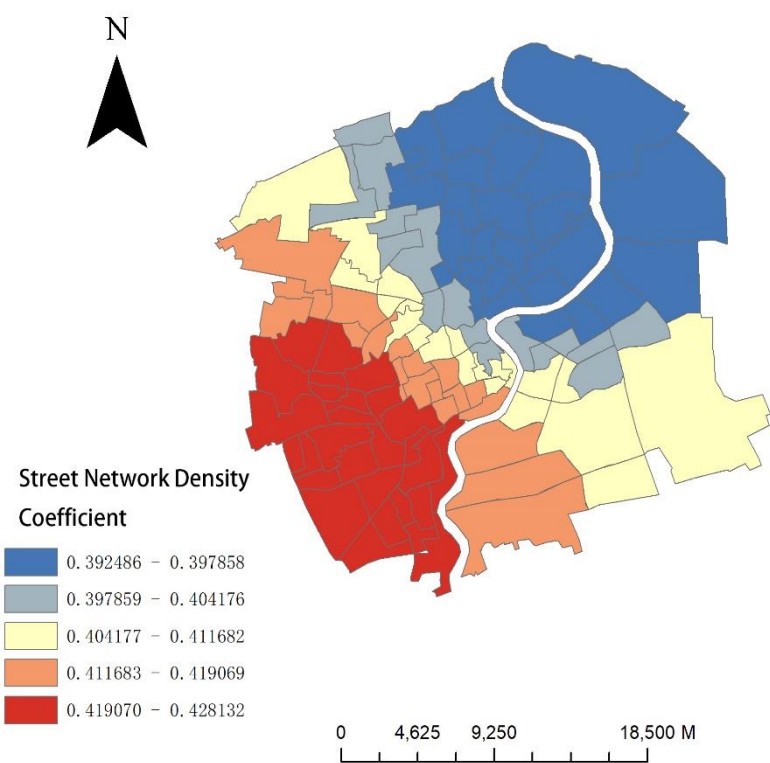

**Figure 14.** Street-network-density coefficients at each county or jiedao.

## 4. Discussion

This study concentrated on street greenery within the outer ring of Shanghai. Compared to urban parks or other kinds of green spaces, researchers may leave street greenery out of consideration despite its close contact with residents in their daily life. Meanwhile, green view as an indicator to evaluate the intuitive perceptions of residents is efficient and effective. Using Baidu Map street images and socioeconomic data, the relationships between different socioeconomic indices and the spatial distribution of street greenery were successfully distinguished. The occurrence of environmental injustice among different districts within the outer ring of Shanghai should be paid more attention to by the local municipality.

### 4.1. Policy Implications

Different from previous studies using high-resolution satellite pictures to calculate green-space coverage, our study adopted street-view pictures to explore residents' intuitive visual perceptions on the surrounding environment, which is more authentic in that street-view photos simulate residents' daily access to the environment, including buildings, street greenery, and vehicles [29]. While urban green canopies derived from remote-sensing methods such as satellite images could quantify urban green spaces [49], they fail to evaluate people's sensory functions. In addition, this kind of intuitive perception of residents, GVI, also takes the human-oriented perspective into consideration, giving a novel comprehension compared to former studies based on the accessibility of different social groups to urban parks [23,28,50]. An indicator based on residents' demand-side perceptions is used to replace the supply side accessibility calculated for urban green spaces. For instance, even though the urban green coverage of Shanghai equals to 39.6% [51] which nearly reaches the ecological environmental standard of 40% urban green coverage of international metropoles suggested by WHO and Chinese National Forestry and Grassland Administration [52], green-view values in reality are inconsistent with the noticeable urban green coverage based on this study. In other words, the local municipality only concentrated on the supply side indicators of urban green coverage from a macro-city

level, but without considering residents' real demands of feeling greenery. This may have been caused by the preference of urban-green-space construction in China, namely, local municipalities tend to construct gated city parks in all districts and plan large amounts of wetland and nature reserves in rural areas, but leave the accessibility of different social groups out of consideration. Consequently, such spatial layouts of green spaces may result in residents who want to access green spaces having travelling costs, since gated urban parks often cluster in different subcenters or rural–urban areas. Hence, despite the quantity of urban green coverage of Shanghai being up to the standard suggested by the WHO and the Chinese government, the majority of green spaces are less accessible than street greenery is for local residents. For example, people with restricted leisure time arising from longer working hours and lower incomes have less of a chance to access urban parks and green spaces, although their residences are closer to urban vegetation than other communities are [53]. Therefore, local municipal policies of green-space construction should combine the supply side indicator of urban green coverage and the demand-side indicator of residents' perceptions.

Compared to the macroscale of prefecture-level cities in the study of Chen et al. on urban greening quality in the Pearl River Delta urban agglomeration (PRDUA) [38], this study focused on a more micro scale at the county level, which might have drawn a more accurate conclusion of the relationship between socioeconomic and built-environment indices, and the spatial distribution of street greenery. Meanwhile, the study area covering the entire downtown area of Shanghai also provided sufficient data and samples. With regard to the aggregated values of GVI, this study adopted the Thiessen polygon method to mitigate the bias of densely located sampling points, which gives a more accurate result than the simplistic mean value of sampling points in the study of Chen et al. While Long and Liu massively calculated the average GVI of the 131 prefecture-level cities in China [54], this study more comprehensively analyzed how social status and the built environment influence the spatial layout of GVI and the possible environmental injustice in Shanghai. Factors correlated to GVI selected in parts of studies might be oversimplified. For instance, Wang et al. merely took housing price and population density as the key factors [55]; Li et al. only focused on the relationships between census data and the average GVI [34]. A mixture of both the socioeconomic and physical-environment indices adopted in this study may be more precise and reasonable, since both result in injustice in the distribution of urban greenery.

Our study identified the spatial agglomeration of the distribution of street greenery, indicating that there might be environmental injustice for residents to feel green spaces. For instance, there exists strong positive correlation between aggregated GVI and the built environment. The positive correlation indicates that a more upscale district or county equipped with a high density of walkable streets benefits from a high ratio of green view. The high density of street networks, especially footways and walkable streets, are usually planted with more trees or brushes, providing more visible greenery for residents. In addition, high GVI values at counties or jiedaos in the southern and western parts of Shanghai are more correlated with the high density of street networks. Such a finding can also provide strategic implications for current urban regeneration in Shanghai. During the process of regeneration, the local municipality should encourage and construct a more walkable street network in communities with more visible greenery throughout the central urban area to more evenly distribute plant resources. The strong positive correlation between district or county with more high-end communities and the high GVI value is consistent with some previous studies [38,55]. In this study, the positive correlation between GVI value and housing price was more significant in the northeast of Shanghai, where there are a number of new upscale communities. On the other hand, the local municipality should be conscious of the negative correlation between aggregated GVI and the ratio of vulnerable groups. Children younger than 16 years old and people older than 60 years old need to have improved exposure to visible urban greenery since they can become more physically and psychologically healthy through physical and social activities

when they can access green space [56]. Such negative correlation is more significant in the central urban area than that in peri urban areas; therefore, policy makers should focus more on these counties or jiedaos located in central urban areas with a high ratio of the elderly and children when developing green spaces. On the basis of the results mentioned above, urban planners and policy makers could propose a more just distribution of visible green spaces for residents, to meet the demands of human-oriented development and narrow the gap of environmental rights for different social groups.

### 4.2. Limitations

When interpreting our results, the limitations of the study, especially ones related with sampling, should be considered. The GVI method, which crawled street-view photos from the Baidu Map, has some uncertainties such as seasonal impact on greenery and new image-recognition technology. Street greenery widely varies in different seasons in Shanghai, especially summer and winter, since plants in Shanghai are usually composed of deciduous trees that present a lower GVI at winter. Air pollution and foggy days at winter may also influence the green-view acquisition and worsen GVI results. As a result, improvements on crawling street-view photos in different seasons should be considered to compare the GVI in different seasons or even in real time.

In addition, with the purpose of improving accuracy, more advanced AI-based image-recognition algorithms could be introduced to precisely recognize plants and trees, and to mitigate the intrusions of other nonbotanical green items in street-view photos. The algorithm in this study adopted the HSV color range to recognize greenness in street landscapes, which may incorrectly include nonbotanical green items. Hence, except for the color of the street landscape, all objectives including the different species of trees and greenness could be examined to analyze the relationship between botanical species and GVI through an advanced quantitative AI-based algorithm.

### 5. Conclusions

This study adopted green-view values as an indicator to evaluate the street greenery of the downtown area of Shanghai based on Baidu Map street-view images. GVI values of the majority of the sampling sites, which cover the entire downtown area of Shanghai, were lower than 30%—namely, the value of a comfortable visual environment proposed by Aoki [36]. More street greenery could be seen in the inner ring, especially the Xuhui and Huangpu districts. With regard to the influential factors of the green view, housing price, POI diversity, and street-network density positively influence GVI values on the county level. On the other hand, GVI values were negatively impacted by the ratio of vulnerable groups. Population density did not impact the spatial patterns of the green view. Hence, environmental injustice according to the correlations between GVI values and socioeconomic and built-environment factors should be considered by the local municipality. This indicates that prosperous districts with a completed design usually possess more human-oriented green spaces compared to those with a dilapidated living environment. Less street vegetation could also be seen at counties and jiedaos with more vulnerable groups. Concerning street greenery as a kind of public resource, policy makers and urban planners should concentrate more on balancing the spatial distribution of street greenery through the evaluation of GVI values in the downtown area of Shanghai to mitigate possible environmental injustices. Future studies may focus on how different tree species influence the GV values on a detailed community scale to obtain a more accurate result.

**Supplementary Materials:** The following are available online at https://www.mdpi.com/article/10.3390/land10080871/s1, Supplementary Materials: GWR Result.

**Author Contributions:** Conceptualization, software, data curation, and writing—original draft, C.-J.G.; methodology, writing—review and editing, validation, and funding acquisition, C.X.; supervision and discussion, Q.S.; All authors have read and agreed to the published version of the manuscript.

**Funding:** This work was supported by the National Natural Science Foundation of China (grant number: 72072131) and the National Social Science Fund of China (grant number: 20CGL047). Any opinions, findings, and conclusions or recommendations expressed in this material are those of the authors and do not necessarily reflect the views of the National Science Foundation.

**Institutional Review Board Statement:** Not applicable.

**Informed Consent Statement:** Not applicable.

**Data Availability Statement:** The presented data are shown in the paper, more details are available from the corresponding author.

**Conflicts of Interest:** The authors declare that they have no known competing financial interests or personal relationships that could have influenced the work in this paper.

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
