# Peer review of "Assessing the Spatial Distribution Pattern of Street Greenery and Its Relationship with Socioeconomic Status and the Built Environment in Shanghai, China"

_land, doi:10.3390/land10080871_

Round 1

Reviewer 1 Report

This is a very interesting and original paper with important policy implications, not just in China, but more widely. The methodology is clear, and the conclusions are supported by the results. However, the English grammar needs major revision by a native English speaker before it can be published.

The paper assesses the equity of the spatial distribution of street greenery using a novel approach – green view - which uses street-level data rather than satellite images, thereby enabling a perspective on urban greening that is more relevant to how people experience city streets. Focusing on the microscale and overlaying socio-economic data reveals the spatial inequity of street greenery in Shanghai. The methodology is clearly explained such that it can be replicated by others, and the conclusions are consistent with the evidence and arguments presented. On the whole the paper is well written, but the manuscript needs to be revised by a native English speaker, as there are numerous grammatical errors, so it cannot be published in its present form. 

Author Response

Revision Note

Journal: Land

Manuscript ID: land-1296204

Title: Assessing Equity of the Spatial Distribution Pattern of Street Greenery and its Relation with Socioeconomic Status and Built Environment in Shanghai, China

Revised Title: Assessing the Spatial Distribution Pattern of Street Greenery and its Relation with Socioeconomic Status and Built Environment in Shanghai, China

Authors: Chao Xiao, Qian Shi and Chen-Jie Gu*

The authors sincerely appreciate the constructive comments by reviewers, and the revised manuscript had been revised by the English editing service of MDPI. The text has been checked for correct use of grammar and common technical terms, and edited to a level suitable for reporting research in a scholarly journal.

Reviewer 2 Report

In my opinion the value of the presented article is the assessment of street greenery in the view from the human level. Such studies are sporadic and bring a lot of value to the evaluation of greenery in cities.

However, I have comments on the research presentation and the article needs significant improvement.

There should be more reference to worldwide studies in the introduction. .  At the moment the article has a local character.  The purpose is described, but the research questions are missing,

Methodology

The method is described too vaguely and requires a more precise description. Too poorly described method of selecting points for GV assessment and their assessment. Whether they were all streets or only selected streets. Fig. 1 Too small and illegible, no legend. Fig. 2 Illegible.

The description of the method in Line 118-120 - unclear, requires a precise description.

Description of the method 151-158 - requires a more precise explanation.

Reference to inner and outer ring appears in text - no explanation of this concept (e.g. Line109, 208, 226, 227, 231, 232, 235, 276, 2770 .

Fig. 3 No boundaries between the 4 images - a, b, c, d are shown. Their size is not known

Results.

In this part there are elements that should be in Discussion eg. Line 200-208, line 260-281.

Fig. 7 Illegible, it is impossible to read places on the map according to the legend (no colors)

Correlation between GV and other variables on Fig. 11 requires a more detailed description. It is currently unreadable.

Discussion needs thorough improvement. The results should be discussed with the research of other researchers in relation to the research questions posed, which are currently missing.

Author Response

Thanks for the comments. Revised manuscript has been uploaded. Please see the attachment for a point-by-point response.

Reviewer 3 Report

This paper examines the relationship between the abundance of vegetation from a human perspective, measured with an index named Green View, and various socio-economic factors in Shanghai. The paper is clearly structured and tries to address an important issue in urban planning. However, it needs further specifications, particularly about its contribution to the literature and measurement methods for the variables, in order for this manuscript to be considered for publication. Moreover, it is worth conducting a multivariate analysis instead of the simple correlation analysis to consider confounders and solidify the arguments.

My major comments are the following:

  1. The contribution of this paper to the literature was not clear for me. According to the last paragraph in the introduction, this paper aimed at (1) proposing “a viable and intuitive method to calculate the street greenery from residents’ perception” and (2) exploring the distribution of the green view and that of other socio-demographic variables, as well as their relationships. However, as for the first point, the proposed metric Green View (GV) seems highly similar to an established index called Green View Index (GVI), which has been widely used in the literature (e.g., Li et al. 2015 and Yang et al. 2009). What exactly is the difference between GVI and GV? Is the difference so fundamental that the proposed GV should have a different name from GVI? The explanation in the current manuscript was not convincing enough.

  1. As for the second point, the motivation for studying environmental injustice in this paper was not explicitly explained. It is well-known that there is a disparity in the distribution of urban vegetation and that of various characteristics in a neighborhood (e.g., Boone et al. 2009, Li et al. 2016), and the authors cite several articles concerning environmental injustice. However, the missing piece in the literature was not explicitly pinpointed in the current manuscript, thereby failing to claim the need of studying the injustice/inequality of urban greenery. What remains unknown (and worth understood) in the literature, and how the point can be addressed?

  1. Also for the second point, the analytical methodology (correlation analysis) is not solid enough to claim the presence of “patterns” in the variables, because the observed correlations might be spurious. Further statistical analysis such as multivariate regressions is highly recommended to consider the interactions between the variables and make a more credible statement about the injustice of Green View.

  1. How were the sampling places chosen? The explanation “arbitrarily place the sampling points with an interval of 100 m alongside the streets” (line 116-117) is not clear enough: does a street segment of a length less than 100 m not have any sampling point on it at all?

  1. How to justify the aggregation of point-based GV value to the county level by simply taking the mean for the sampling points that are located in a county? A study demonstrated that such a simplistic aggregation includes bias from the location of sampling points and the street network structure itself (Kumakoshi et al. 2020). I am not convinced that the aggregated GV values at the county level are an unbiased measure and that the discussion built on it is robust.

Some minor comments are the following:

Shanghai was chosen for the study area because it “reflect the common phenomenon of street greenery in many mega cities around the world” (line 100-101) but a part of the obtained result in Shanghai was inconsistent with previous studies (e.g., line 280-282). What exactly is “the common phenomenon” and why the obtained result has some inconsistency with previous studies?

Concerning the studies using NDVI, the spatial resolution of satellite images is usually as high as 1 m square or 30-50 cm square, which should be highly accurate. What did the authors mean by stating “Some drawbacks exist within such studies including the accuracy of data at district scale, and the rationality of the ‘accessibility’”? (line 62)

How was the road network density (line 179) measured? What are the denominator and the nominator?

It would be nice to put the scale in meters in Figure 1 and Figure 7 because the current manuscript expresses the distance in meters. Also, Figure 7 would be more understandable if the color scale is modified: it is hard to tell from the current image whether a district has actually full of green vegetation or just it has a lot of sampling points.

As for the measurement of spatial patterns, the authors stated that there is a certain agglomeration (autocorrelation) of counties with high GV values because Moran’s I score is 0.08 (p<0.01) (line 222). What was used as the matrix of spatial weights? I think this piece of information is necessary when using Moran’s I.

The interpretation of the results was discussed in section 3.2 (particularly lines 228-237) combined with the land use of the study area. However, I could not understand where the land use information came from. For those who do not know Shanghai, the sentences in this part lack evidence.

I would recommend asking a native speaker of English to review the manuscript, because some expressions seemed to need correction or specification, such as “the spatial distribution of urban greenery is injustice” (line 47), “this study tends to analyze the correlation between… “ (line 246), and “residents who want to be access to green space…” (line 313-314).

References

Boone, C. G., Buckley, G. L., Grove, J. M., & Sister, C. (2009). Parks and people: An environmental justice inquiry in Baltimore, Maryland. Annals of the Association of American Geographers99(4), 767-787.

Kumakoshi, Y., Chan, S. Y., Koizumi, H., Li, X., & Yoshimura, Y. (2020). Standardized green view index and quantification of different metrics of urban green vegetation. Sustainability, 12(18), 7434.

Li, X., Zhang, C., Li, W., Ricard, R., Meng, Q., & Zhang, W. (2015). Assessing street-level urban greenery using Google Street View and a modified green view index. Urban Forestry & Urban Greening, 14(3), 675-685.

Li, X., Zhang, C., Li, W., & Kuzovkina, Y. A. (2016). Environmental inequities in terms of different types of urban greenery in Hartford, Connecticut. Urban Forestry & Urban Greening, 18, 163-172.

Yang, J., Zhao, L., Mcbride, J., & Gong, P. (2009). Can you see green? Assessing the visibility of urban forests in cities. Landscape and Urban Planning, 91(2), 97-104.

Author Response

Thanks for the comments. The revised manuscript has been uploaded. Please see the attachment for a point-by-point response.

Reviewer 4 Report

Dear Authors,

The manuscript "Assessing Equity of the Spatial Distribution Pattern of Street Greenery and its Relation with Socioeconomic Status and Built Environment in Shanghai, China" is promising research to analyze and assess urban green space towards improving human-wellbeing. I think this could be a very valuable contribution to urban green space planning for LAND. However, the current form of the manuscript requires major revision. 

Firstly, as written "this study explored the spatial distribution of street greenery and its influential factors using green view (GV) as the main evaluation indicator". 

I feel the framing of your manuscript, i.e. focusing on equity and injustice, is a weakness. I am not saying these aspects are not important. I would like to see the introduction re-written using a framework that promotes the need for GV in urban planning of green space for human well-being.  Focus more on improving urban green space planning methods. Don't focus on social injustice or equity in the introduction .

The injustice and equity should come in the discussion based on your results if you want to keep. For instance this study shows the spatial distribution and availability to society varies throughout Shanghai.  Then You did find a variety of important results e.g. rich areas had better GI. Green space was lacking in poor social-economic areas, therefore green space planning is urgently needed in these poorer areas. This may also indicate that green space planning has focused on richer suburbs which then may imply social injustice.  I also propose removing equity from the title to make simpler and stand out more. 

Secondly, the manuscript need to be edited by a native English speaker or somebody with proficient English skills, there are just too many mistakes which makes the manuscript difficult to read and understand.  This limits the take home message of your interesting scientific study.

Thirdly, the figures can be improved including the figure text. Don't be afraid to explain what is in the figure. The caption text should be informative. The figure are generally very small see figures 1-3. Indeed this could be made in to on nice figure. What is the difference between figure 1 and figure 7? Figure seven has some values but these are not explained. Figure 5 should be a table not a figure. Please revise your figures, good figures can really help the readers.

Fourthly, I thing you can improved the conclusion. I think you can conclude that the use of green viewer would help policy makes and planners make informed decision is the urban planning of green space for human well-being. But you have seemed to missed this point.

Also I thing the referencing needs improving. Please check LAND's guidelines reference should be numbered i.e. [1] not (2019) as found in some places. Also I would suggest reading and adding the following paper into your manuscript  https://www.mdpi.com/1999-4907/11/12/1347. This study is similar but focuses on Evaluating street greenery in Nanjing, China.

Attached you will find the manuscript with many comments and proposed edits.

Thank you and keep up your important research. I trust my review and suggestion will help you to improve your manuscript. 

All the best.

Author Response

Thanks for the comments. The revised manuscript has been uploaded. The text has been checked for correct use of grammar and common technical terms, and edited to a level suitable for reporting research in a scholarly journal. Please see the attachment for a point-by-point response.

Round 2

Reviewer 2 Report

The manuscript has been revised according to my suggestions. I accept it in its present form

Author Response

Thanks for the reply. We used the standardzied coefficient of the variables in the table and figures to improve the comparison of coefficients of the models. Also, we revised the abstract and some content in the result part. Please see the revised manuscript.

Reviewer 3 Report

Thank you for all the revisions and your efforts to investigate the issues that I raised in the first review. It seems that the paper improved the soundness of the argument, particularly by clarifying the measurement methods of the variables and adopting multivariate regression models.

However, there are still some issues that I think should be solved. I consider these points crucially important to guarantee the quality of this paper as an academic article, which are (1) description of this study’s motivation and originality in a consistent form and (2) appropriate representation of the analytical results and their interpretation.

Concerning the issue (1), the authors added a sufficient amount of previous works related to the current study (e.g., response to comment 2) and seemingly concluded that the environmental disparity has not been discussed at the county or community level (l.75-77). If this is true, the existence of this gap in the literature should be emphasized in the abstract as well: the current form of the abstract seems to put too much emphasis on the characteristics of GVIs, without referring to the knowledge gap in the literature of environmental injustice. I feel that the current form of the abstract does not fully describe the originality of this study.

Additionally, the paragraph in l.322-l.332 (e.g., response to comment 5) was not clear; I expected that the interpretation in this paragraph was clarifying the particularity of the study area, but the use of some new words made it difficult to grasp the authors’ interpretation. What exactly was the relationship between the population density (at the county level I suppose?) and the presence of greenery measured by GVI? What does it mean “central Shanghai areas with large population density are evenly distributed in the inner ring” (l.325-326; I couldn’t understand what “areas” meant in relation to county/community/jiedao)? And what does it mean “population density is uncorrelated with GVI values (sig > 0.05), which is widely recognized as a positive influence on GVI values” (l.323-325; the influence of what to what?)? Please clearly state your interpretation.

As for issue (2), using OLS and GWR seems to fit the objective of this study, namely studying the environmental injustice in geographical terms. However, the simple comparison of coefficients of the models (e.g., l.340-342) is not appropriate unless the unit of the variables is fully considered because the coefficients are highly sensitive to the variable units. For instance, an increase in one point of street network density (in meter, according to the response to comment 6) and that of housing price (yuan?) may not have equal importance. To compare the influence of each variable to the dependent variable, I would recommend either using standardized coefficients or modifying the variable units (i.e., divide the housing price by 100 yuan).

I would also suggest modifying the following points in the analytical part:

  • Describe the confidence intervals of the estimated coefficients in the OLS and GWR models because t value needs thresholds and it is somewhat cumbersome for readers to interpret the statistical meaning of the estimation.
  • Clearly state where the information in Figure 12-14 is from: are these coefficients in the GWR model?

Author Response

Thanks for the reply. We used the standardzied coefficient of the variables in the table and figures to improve the comparison of coefficients of the models. Also, we revised the abstract and some content in the result part. Please see the revised manuscript and the attachment.
